# Group-specific archaeological signatures of stone tool use in wild macaques

Lydia V Luncz[1]*, Mike Gill[1], Tomos Proffitt[2], Magdalena S Svensson[3], Lars Kulik[4], Suchinda Malaivijitnond[5,6]

[1]Primate Models for Behavioural Evolution Lab, University of Oxford, Oxford, United Kingdom; [2]The Institute of Archaeology, University College London, London, United Kingdom; [3]Department of Social Sciences, Oxford Brookes University, Oxford, United Kingdom; [4]Department of Primatology, Max Planck Institute for Evolutionary Anthropology, Leipzig, Germany; [5]Faculty of Science, Chulalongkorn University, Bangkok, Thailand; [6]National Primate Research Centre of Thailand, Chulalongkorn University, Saraburi, Thailand

**Abstract** Stone tools in the prehistoric record are the most abundant source of evidence for understanding early hominin technological and cultural variation. The field of primate archaeology is well placed to improve our scientific knowledge by using the tool behaviours of living primates as models to test hypotheses related to the adoption of tools by early stone-age hominins. Previously we have shown that diversity in stone tool behaviour between neighbouring groups of long-tailed macaques (*Macaca-fascicularis*) could be explained by ecological and environmental circumstances (Luncz et al., 2017b). Here however, we report archaeological evidence, which shows that the selection and reuse of tools cannot entirely be explained by ecological diversity. These results suggest that tool-use may develop differently within species of old-world monkeys, and that the evidence of material culture can differ within the same timeframe at local geographic scales and in spite of shared environmental and ecological settings.

DOI: https://doi.org/10.7554/eLife.46961.001

*For correspondence:
Lydia.Luncz@anthro.ox.ac.uk

**Competing interests:** The authors declare that no competing interests exist.

## Introduction

The cultural evolution of early hominins is investigated through the identification of archaeological evidence, primarily in the form of stone artefacts and bones (*Leakey, 1971*; *Schick and Toth, 1994*). The emergence and evolution of tool use is generally accepted as a transformational process that led to a suite of adaptations and behaviours, enabling humans to become the most successful primate on the planet (*Hovers, 2015*). The earliest production of deliberately flaked stone tools dates to ~3.3 Mya (*Harmand et al., 2015*; *Lewis and Harmand, 2016*). It has, however, been hypothesized that simple pounding tools may have been a precursor to intentionally flaked technology, potentially originating around or before the period of our last common ancestor with chimpanzees (*deBeaune, 2004*; *Marchant and McGrew, 2005*; *Rolian and Carvalho, 2017*; *Thompson et al., 2019*).

Although rare, percussive stone tool use is also practiced by a few species of non-human primates. These include Western chimpanzees, *Pan troglodytes verus*, (*Boesch and Boesch, 1990*; *Sugiyama and Koman, 1979*), bearded capuchin monkeys, *Sapajus libidinosus*, (*Fragaszy et al., 2004*; *Ottoni and Izar, 2008*; *Falótico and Ottoni, 2016*; *Luncz et al., 2016a*), long-tailed macaques *Macaca fascicularis*, (*Malaivijitnond et al., 2007*; *Gumert et al., 2009*; *Luncz et al., 2017a*) and one group of white faced capuchin monkeys, *Cebus capucinus imitator* (*Barrett et al., 2018*). These non-human primates, similarly to hominins, create long lasting archaeological signatures comprised of lithic assemblages (*Mercader et al., 2002*; *Mercader et al., 2007*; *Proffitt et al., 2018*;

*Falótico et al., 2019*). By merging the fields of archaeology and primatology it is possible to combine direct behavioural observations with the archaeological signature of tool use (*Haslam et al., 2009*; *Rolian and Carvalho, 2017*; *Carvalho and Almeida-Warren, 2019*). Lithic assemblages remain the most abundant source of evidence for understanding early hominin technological and cultural variation (*Delagnes and Roche, 2005*), as well as social learning (*Stout et al., 2019*). As such, the field of primate archaeology is well placed to help investigate variation of tool evidence and cultural evolution. Further, this new research field is providing the ability to test various archaeological hypotheses often applied to the Early Stone Age (ESA) archaeological record. Examples of such empirical work include the effect that distance from raw material source has on lithic reduction (*Braun et al., 2008a*; *Luncz et al., 2016b*); the effect of population size on resource depletion (*Steele and Klein, 2005*; *Luncz et al., 2017b*); and chaine-operatoire approaches to stone tools (*Delagnes and Roche, 2005*; *Carvalho et al., 2008*). Archaeological techniques applied to the primate record have also been used to identify evidence of past tool behaviours and expand the temporal signature of tool use in non-human primates (*Mercader et al., 2002*; *Haslam et al., 2016a*; *Haslam et al., 2016b*; *Luncz et al., 2016b*; *Falótico et al., 2019*). Furthermore, primate archaeology has proven an effective tool for studying the behaviour of wild non-habituated tool-using populations (*Luncz et al., 2017a*). Reconstruction of tool selection patterns in wild chimpanzees has for example, repeatedly revealed detailed group-specific behavioural patterns (*Koops et al., 2015*; *Luncz et al., 2015*; *Pascual-Garrido, 2019*). This research provides the ability to quantify behavioural diversity, allowing the identification of cultural variation between distinct populations.

Our close genetic relatedness to great apes has led to intense, decades-long research into the tool behaviour of chimpanzees (*McGrew, 2010*). More recently, however, attention is being paid to the tool behaviour of monkeys to broaden the comparative context for the evolution and origin of technology. Long-tailed macaques are the only Old-World monkeys who use stone tools in their daily foraging. This behaviour is mainly observed in populations that live along the ocean shores of Southern Thailand and Myanmar where long-tailed macaques use tools primarily to prey on shellfish, including oysters, marine gastropods, crabs and mussels (*Gumert et al., 2009*), as well as sea almonds (*Falótico et al., 2017*) and oil palm nuts (*Luncz et al., 2017a*). Long-tailed macaques, like humans, use stone tools to kill prey and access meat. These foraging behaviours leave distinct damage patterns (use wear) on the hammerstones. Use wear analysis on stone tools combined with direct behavioural observations has been used to associate tool variation with specific prey species (*Haslam et al., 2013*; *Tan et al., 2015*; *Proffitt et al., 2018*). For example, percussive damage on distal pointed ends of the hammerstone has generally been associated with processing rock oysters (*Tan et al., 2015*), whilst central pitting on a flat hammerstone face is associated with marine snail predation (*Gumert et al., 2009*; *Haslam et al., 2013*).

Our recent research published in eLife (*Luncz et al., 2017b*) focused on providing explanations for observed differences in tool selection between two long-tailed macaque populations. These groups live on neighbouring islands (Koram and Nom Sao Island) on the east coast of Thailand, within the Khao Sam Roi Yot National Park. The results from this study showed that both macaque groups used stone tools to process the same species of shellfish, however the two groups selected different sized stone tools (*Luncz et al., 2017b*). After comparing underlying ecological factors, potentially affecting this disparity in tool size selection, our data suggested that the differences in tool selection were related to the varying prey sizes on each island. Differences in prey size were associated with overharvesting and resource depletion through increased predation pressure on one island. Maximising foraging efforts, macaques appear to preferentially target larger prey, leading to a reduction in the available prey populations and resulting in an overabundance of smaller, sexually immature prey specimens on the more heavily populated Koram Island, in turn leading to the use of smaller hammerstones. Conversely, on the less populated neighbouring island of Nom Sao, the prey population remained abundant and significantly larger in size, resulting in the use of larger hammerstones.

The tool-assisted feed-back loop observed amongst the macaque population at Khao Sam Roi Yot National Park has interesting hominin archaeological similarities. The earliest repeated shellfish exploitation is known to have occurred in South Africa during the Middle Stone Age (MSA) and Later

Stone Age (LSA) (*Steele and Klein, 2008*). Chronological variations in shellfish sizes found in archaeological middens from the both MSA and LSA have been used as proxies for hominin population increases. Increased shellfish collection during the LSA compared to the MSA (*Sealy and Galimberti, 2011*) has been used to argue for and against a long and short chronology for the appearance of modern human behaviour (*Mcbrearty and Brooks, 2000*; *Klein, 2008*; *Steele and Klein, 2005*). However, others have also suggested that changing environmental factors may have been a contributing factor in this shellfish size variation (*Sealy and Galimberti, 2011*). The primate archaeological evidence from Khao Sam Roi Yot provides a direct primate analogy to a phenomenon previously observed through the archaeological record, allowing the ability to directly observe and monitor the effect that population size of tool using primates has on the available shellfish population.

This Research Advance paper builds on and expands our previous research to include further investigation into variation in tool selection and use patterns between long-tailed macaques. By comparing tool selection and use wear patterns between two adjacent islands in the Ao Phang Nga National Park in Southern Thailand we present new information on the possible underlying reasons behind tool variation seen in macaques (*Luncz et al., 2017b*). This area of investigation is not only important for understanding inter-group primate tool variation, but also allows testing of expectations derived from archaeological theory to understand possible underlying factors associated with stone tool variation in the ESA. These include the cultural historic approach and the use of typological analysis to document and understand variation between hominin lithic assemblages (*Leakey, 1971*; *Isaac, 1977a*), the application of a palaeoecological approach, where external environmental factors are seen as important drivers of stone tool variation (*Isaac, 1977b*; *Toth, 1982*; *Toth, 1985*; *Braun and Harris, 2003*; *Braun et al., 2010*). And finally, a chaine-operatoire or technological approach, where emphasis is placed on understanding the complete technological production and use of lithic assemblages, with variation seen to some extent as culturally or socially mediated (*Delagnes and Roche, 2005*; *Roche and Texier, 1996*; *Roche et al., 1999*; *de la Torre and Mora, 2005*).

During exploratory pilot fieldwork on the western side of peninsular Thailand in the Andaman Sea, approximately 500 km southeast of the previous study site in Khao Sam Roi Yot National Park, we identified two island dwelling populations in the Ao Phang Nga National Park, at Boi Yai Island and Lobi Bay (~9 km apart; *Figure 1*) who routinely use stone tools to process the same marine resources. Initial findings suggested possible inter-group tool use differences and as such a possible variation in the macaque archaeological record.

By focusing on intensity of use wear as a proxy for tool re-use and by accounting for underlying ecological and environmental factors of both Boi Yai Island and Lobi Bay, we aim to suggest possible drivers for the observed stone tool variation between the two populations. If environmental circumstances are similar and the factors influencing tool selection, such as prey size, raw material availability and prey availability do not differ between islands, the possibility that tool use behaviour in long-tailed macaques is socially learned cannot be dismissed.

## Results

In total, we analysed 115 stone tools combined from both locations (*Table 1*).

The stone tool variation of the macaque populations of Boi Yai Island and Lobi Bay are separated into two categories; use wear by prey and size of tools selected by prey. *Figure 2* shows examples of the distinct differences of use wear intensity of tools used to process rock oysters (*Figure 2B*) between Lobi Bay (*Figure 2A*) and Boi Yai Island (*Figure 2C*). Similarly, *Figure 3* illustrates the profound differences between use wear grading (UWG) ratings for tools used to exploit *Thais bitubercularis* (*Figure 3B*) on Lobi Bay (*Figure 3A*) and Boi Yai Island (*Figure 3C*).

The comparison between UWG revealed differences between the sites. Stone tools used to crack open oysters are more intensively used on Boi Yai Island compared to Lobi Bay (*Figure 4*, Mann-Whitney U test: U = 32, N1 = 10, N2 = 26, p<0.001). Similar results can be found for UWG on stone tools used to crack *Thais bitubercularis* shells (*Figure 4*, Mann-Whitney U test: U = 1.5, N1 = 9, N2 = 12, p<0.001). The three morphologically similar marine snails (*Nerita* spp., *Monodonta Labio* and *Morulla* spp.) however, produced little in terms of inter-group UWG variation of tool intensity (*Figure 4*, Mann-Whitney U test: U = 381.5, N1 = 27, N2 = 29, p=0.832).

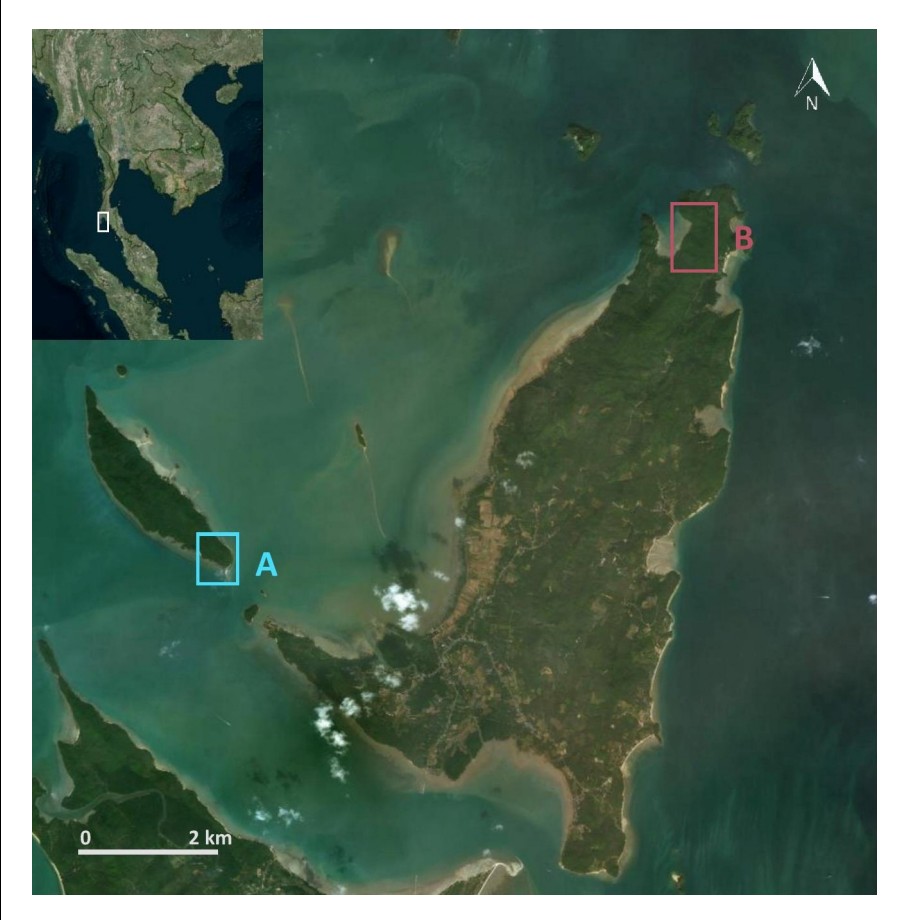

**Figure 1.** Research site in the Ao Phang Nga National Park, Thailand. (**A**) Southeastern tip of Boi Yai Island and (**B**) Lobi Bay on Yao Noi Island, both within Ao Phang Nga National Park, Southern Thailand.

DOI: https://doi.org/10.7554/eLife.46961.002

A comparison of tool size selection associated with prey type revealed differences between Boi Yai Island and Lobi Bay in several important aspects. For marine snails, we found that macaques in Lobi Bay selected heavier tools than on Boi Yai Island (*Figure 5A*, Linear model: E = −1.100 SE = 0.176 $F_{(1,54)}$=38.898 P<0.001). When preying on *Thais bitubercularis* however, both populations selected similarly sized tools (*Figure 5B*, Linear model: E = −0.099 SE = 0.194 $F_{(1,19)}$=0.263 P=0.614). The size of the available prey was similar at both sites (*Figure 6*, site comparison *C. bifasciatus*: Mann-Whitney U test: z = −1.368, N1 = 21, N2 = 73, p=0.086; island comparison marine

**Table 1.** Total number of tools associated with specific prey species.

| Prey | Lobi Bay | Boi Yai Island |
|---|---|---|
| *Monodonta labio* | 9 | 3 |
| *Morulla* spp. | 0 | 2 |
| *Nerita* spp. | 18 | 26 |
| Oysters | 10 | 26 |
| *Thais bitubercularis* | 9 | 12 |
| Total limestone hammerstones used | 46 | 69 |
| Total granite hammerstones used | 0 | 1 |

DOI: https://doi.org/10.7554/eLife.46961.003

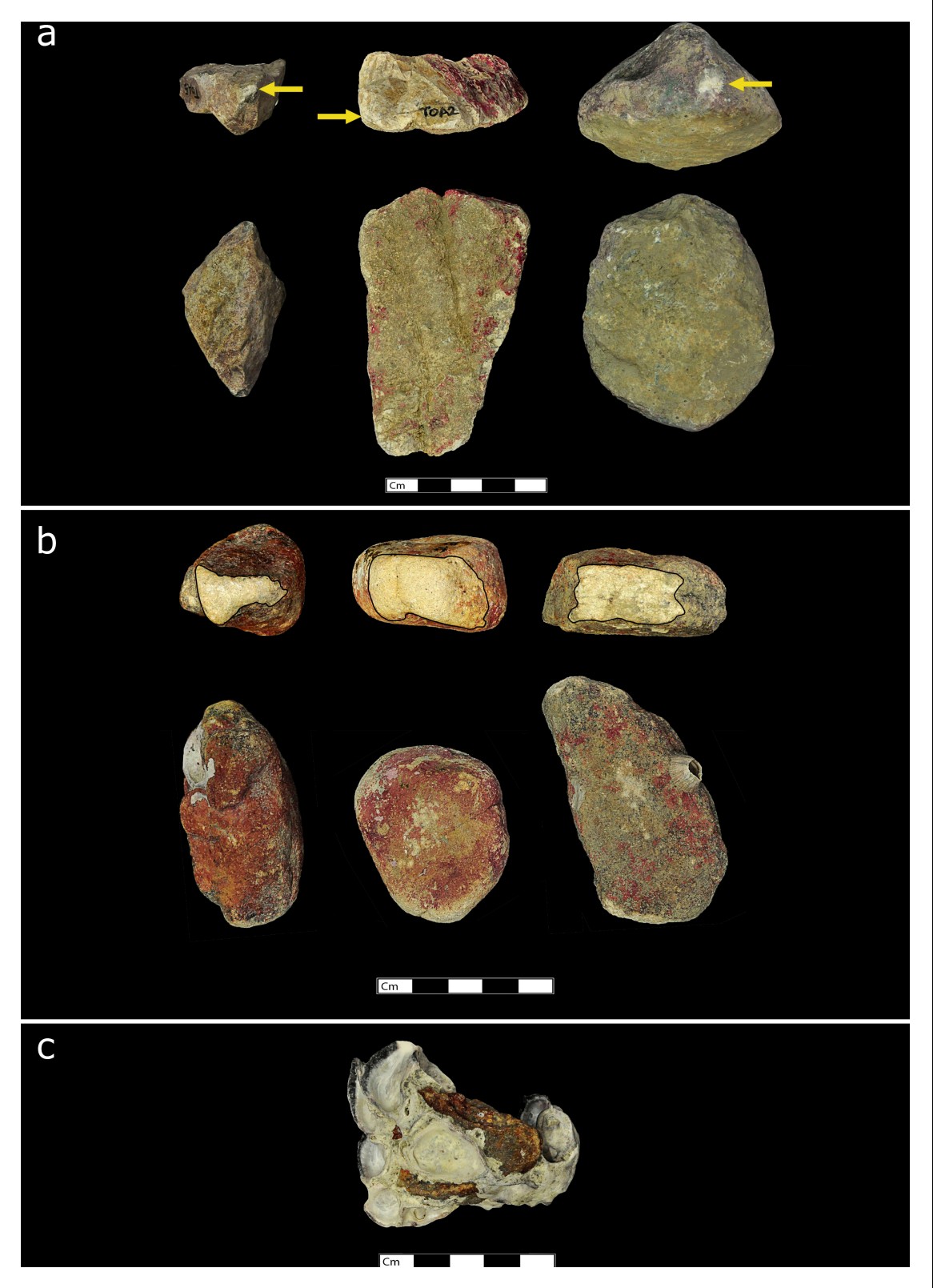

**Figure 2.** Stone tools used by wild macaques in Ao Phang Nga National Park to exploit rock oysters (*Saccostrea cucullate*). (**a**) Examples of stone tools used at Lobi Bay. (**b**) Examples of stone tools used on Boi Yai Island. (**c**) rock oyster prey species available on both islands and.
DOI: https://doi.org/10.7554/eLife.46961.004

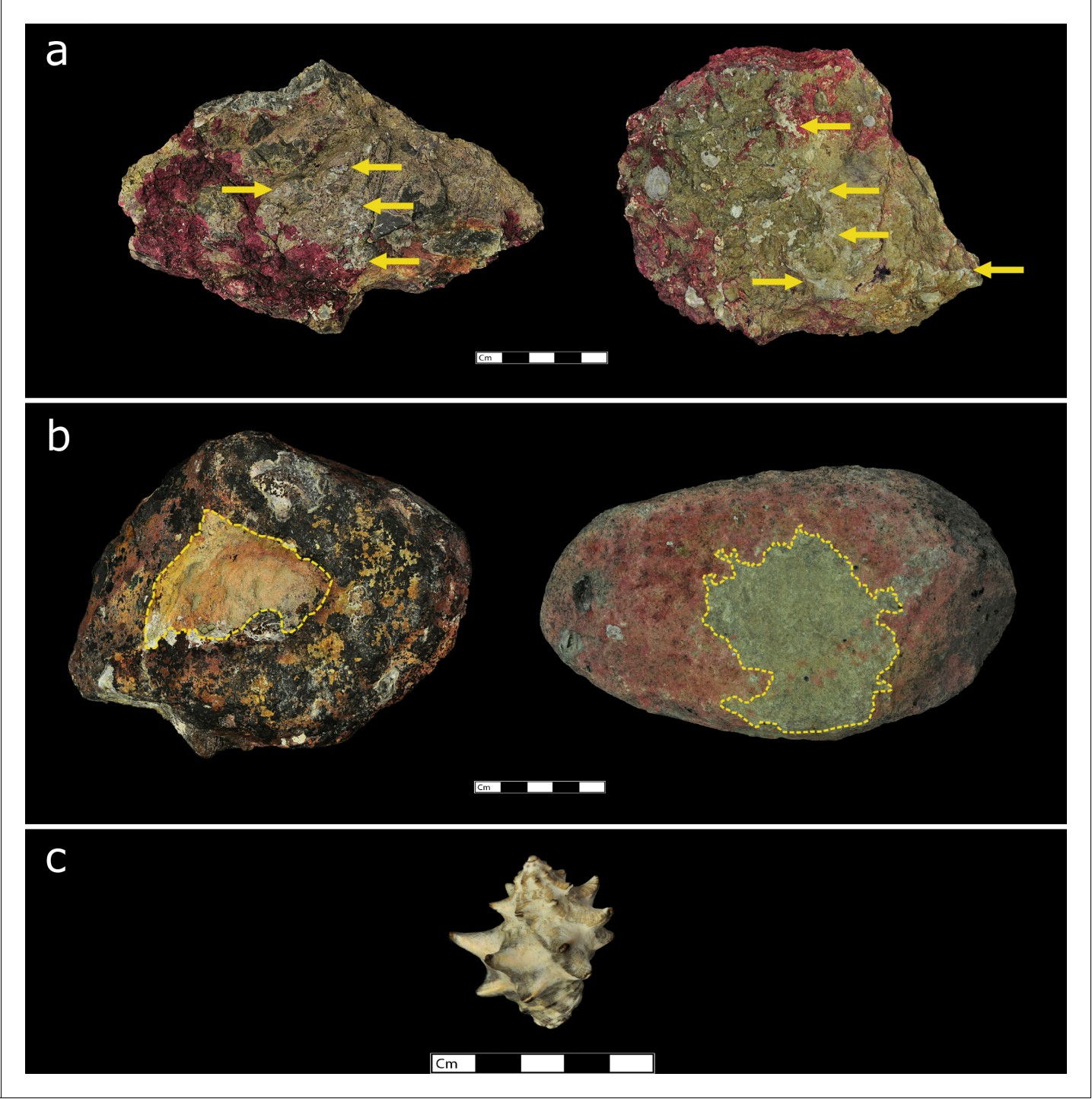

**Figure 3.** Stone tools used by macaques in Ao Phang Nga National Park to exploit *Thais bitubercularis*. (a) Examples of stone tools used at Lobi Bay. (b) examples of stone tools used on Boi Yai Island. (c) *Thais bitubercularis* prey species available on both islands.
DOI: https://doi.org/10.7554/eLife.46961.005

snails: Mann-Whitney U test: z = −0.511, N1 = 123, N2 = 129, p=0.305, island comparison *Thais bitubercularis*: Mann-Whitney U test: z = −0.458, N1 = 14, N2 = 88, p=0.324).

To exploit oysters, however, macaques on Boi Yai Island used heavier tools than those used for processing the same prey in Lobi Bay (*Figure 7*, Linear model: E = 0.451 SE = 0.181 $F_{(1,34)}$=6.197

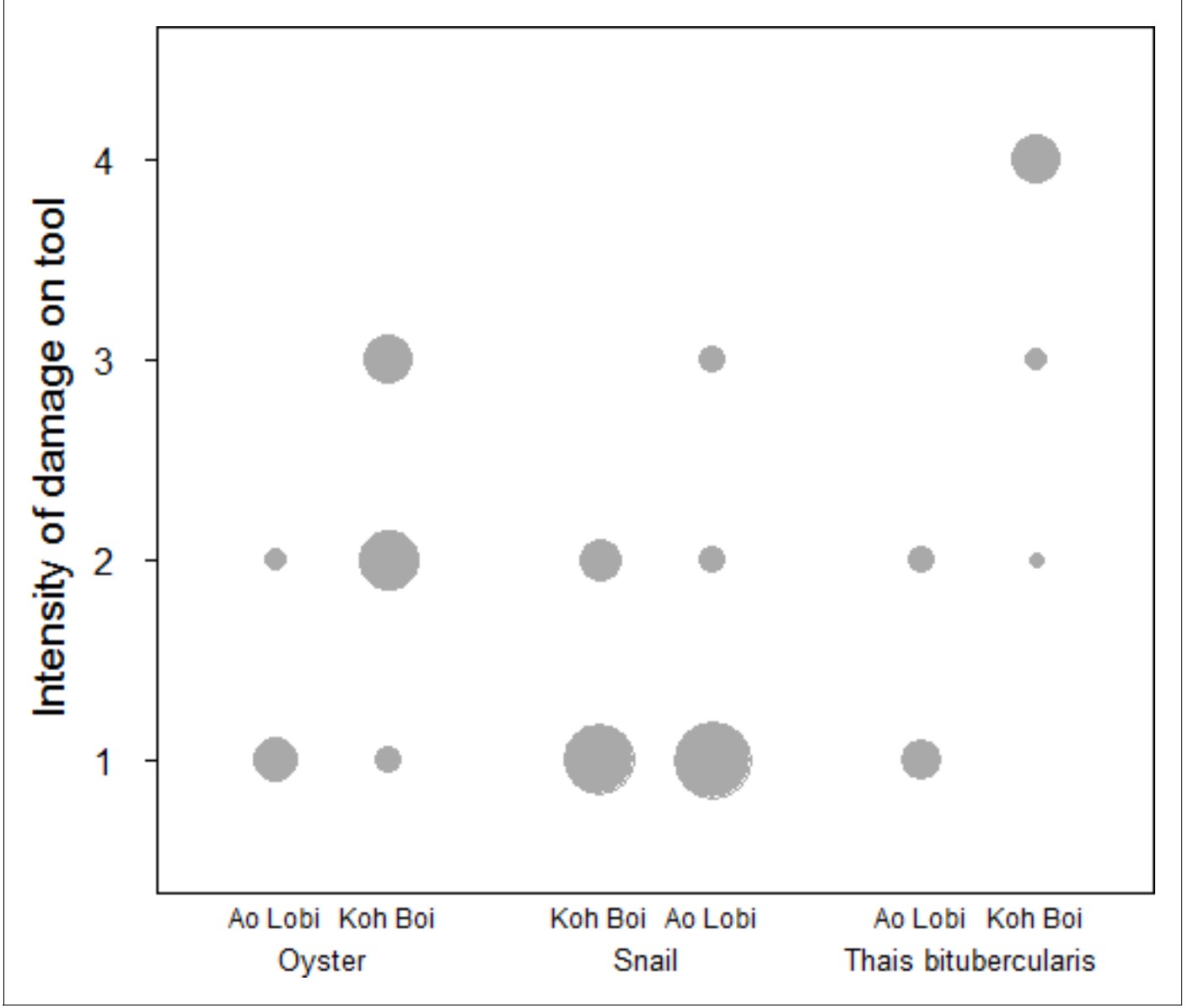

**Figure 4.** Intensity of damage (UWG) compared for stone tools between two sites (Lobi Bay and Boi Yai Island) in the Ao Phang Nga National Park. The size of the circle indicates the respective number of tools included. (For underlying data, see *Source data 1*).
DOI: https://doi.org/10.7554/eLife.46961.006

P=0.018). Oyster size on Boi Yai Island were, in general, larger than in Lobi Bay (*Figure 8*, Mann-Whitney U test: z = −2.862, N1 = 341, N2 = 436, p=0.002).

Several (n = 7) stone tools on Boi Yai Island showed multiple use wear patterns (on the face and on the tip of the tool). Using the methods of *Haslam et al. (2013)*, these damage patterns are indicative of being used on more than one prey species.

Average weight of available raw material (bootstrapped 95% confidence intervals), separated between intertidal and tidal zones differed between sites (*Figure 9A*, Mann-Whitney U test: z = −5.107, N1 = 270, N2 = 432, p<0.001). The raw material availability also differed between sites (bootstrapped 95% confidence intervals over observed plots) (Mann-Whitney U test: z = −1.911, N1 = 20, N2 = 24, p=0.028). However, when separated by tidal and intertidal zones only, availability on Boi Yai Island shows a trend towards more stones available in the intertidal zone (*Figure 9B*,

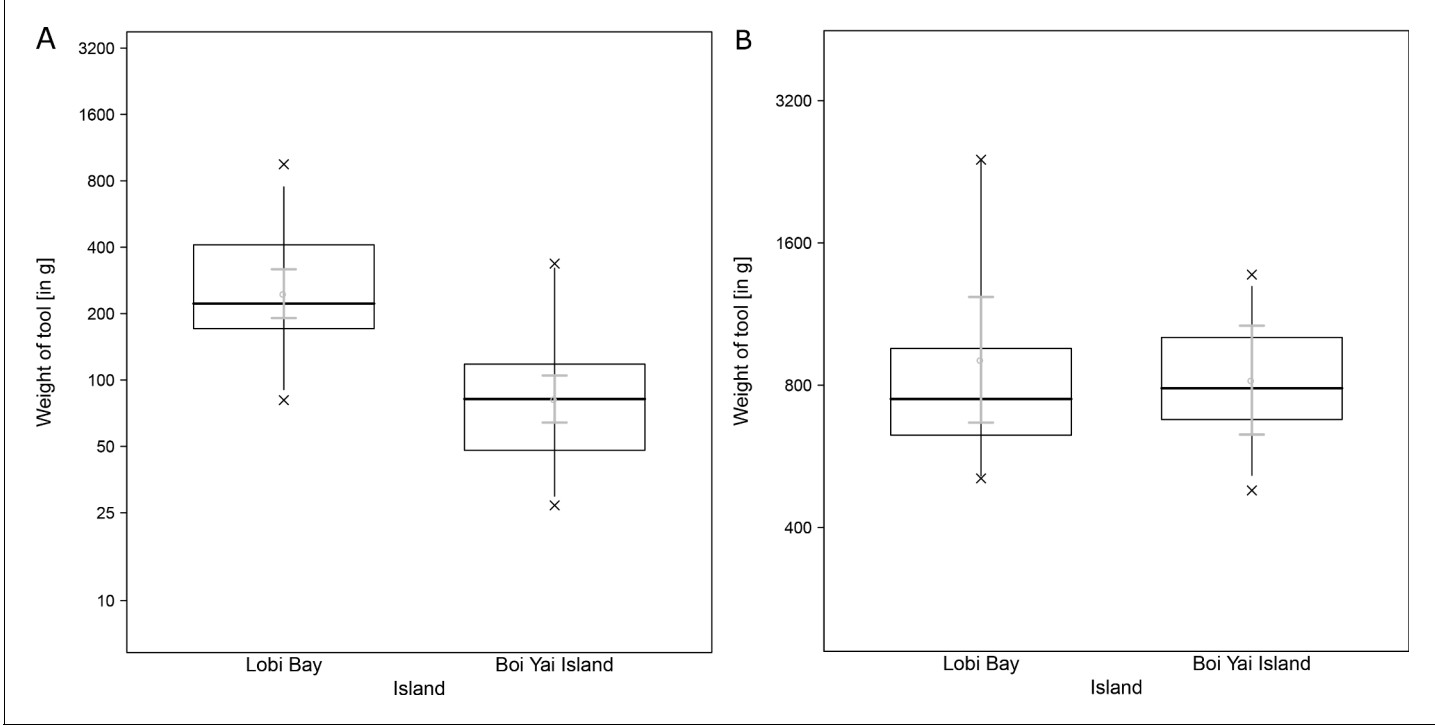

**Figure 5.** Selected tools used to crack open marine prey by wild macaques in Lobi Bay and Boi Yai Island. (**A**) Differences in selected tool weight to crack open marine snails. (**B**) Differences in selected tool weight to crack open *Thais bitubercularis*. The plots are showing all quantiles and the CIs (grey). (For underlying data, see *Source data 1*).

DOI: https://doi.org/10.7554/eLife.46961.007

Mann-Whitney U test: z = −1.374, N1 = 20, N2 = 23, p=0.085). The availability in the tidal zone is similar at both sites (*Figure 9B*, Mann-Whitney U test: z = −1.214, N1 = 20, N2 = 24, p=0.112).

## Discussion

This study explores potential factors behind the variation in percussive stone tool use of two wild macaque populations from adjacent islands in the Ao Phang Nga National Park in Thailand. Macaques on these islands showed differences in their tool selection and degrees of tool reuse when foraging for marine prey. Tool selection was dependent on the species preyed upon. Macaques on Boi Yai Island selected heavier stone tools than those at Lobi Bay to process oysters. Comparisons of available prey species between the sites showed that oysters at Boi Yai Island were indeed larger. The heavier weight of oyster tools used on Boi Yai Island is therefore not surprising and reiterates previous findings, supporting that macaques adjust tool size to prey size (*Luncz et al., 2017b*; *Gumert et al., 2013*). Functional fixedness in regard to stone tool behavioural patterns and target foods can be observed across all tool using non-human primates (*Carvalho et al., 2008*; *Fragaszy et al., 2010*; *Luncz et al., 2016a*). For the larger gastropod (*Thais bitubercularis*) prey, both macaque populations selected similar sized tools to exploit similar sized resources. However, this pattern differs in respect to the processing of marine snails, where the Lobi Bay population selected heavier tools to exploit prey of a similar size to that recorded at Boi Yai Island. Variation in available raw material does not explain the differences in stone tool selection, as the mean weight of available raw material was in fact greater on Boi Yai Island. Furthermore, suitable raw material was available in greater quantities along the shore (especially in the intertidal zone) on Boi Yai Island. Based on this evidence, it is possible to suggest that purely ecological or environmental factors only account for some of the stone tool variation observed between these populations.

When assessing use wear intensity across the two sites, however, additional factors must be explored to explain the apparent variation. Compared to Lobi Bay, the Boi Yai Island tools showed a

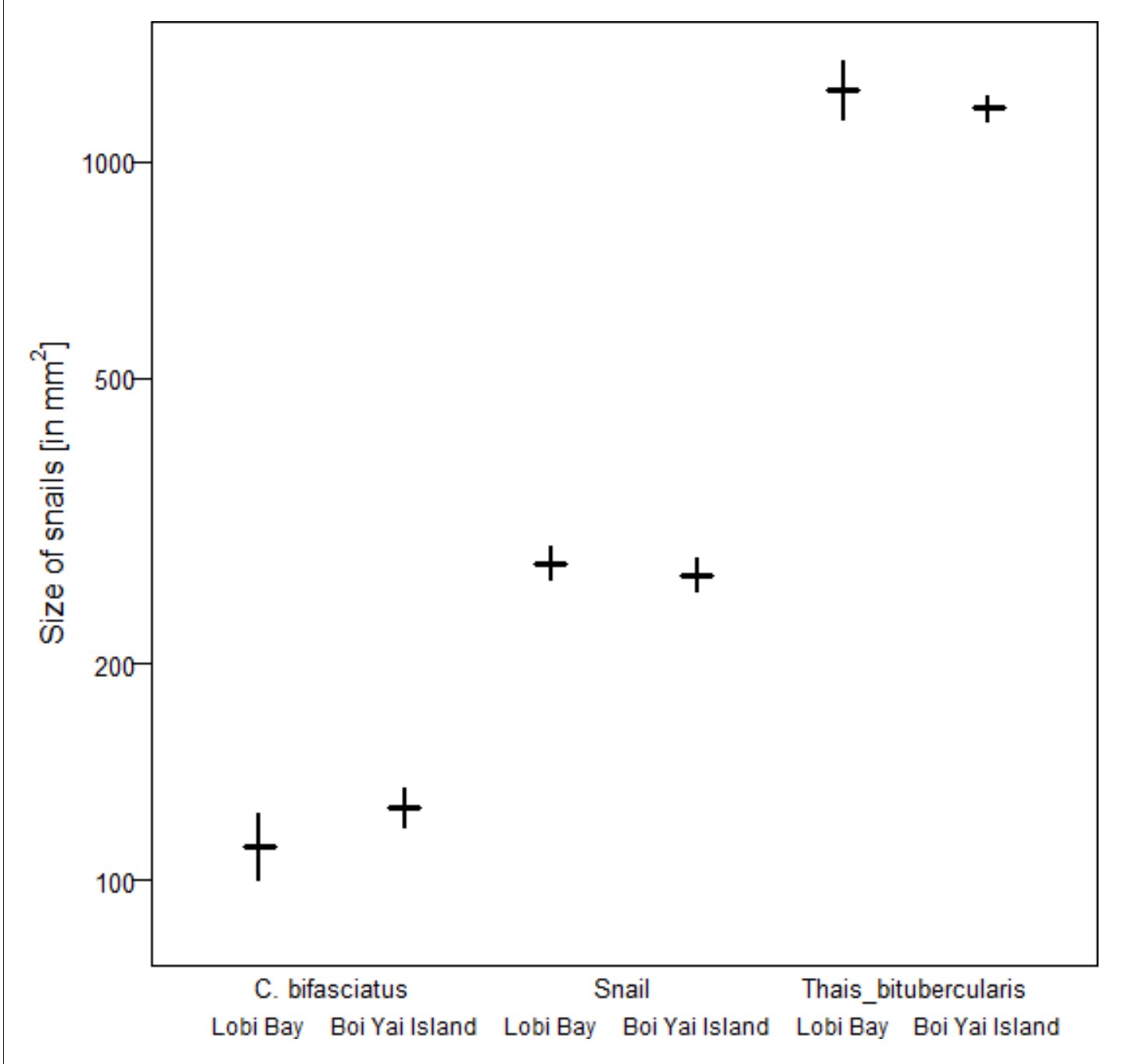

**Figure 6.** Size of marine prey available at Lobi Bay and Boi Yai Island. The plot is showing the mean with bootstrapped 95% confidence intervals.

DOI: https://doi.org/10.7554/eLife.46961.008

The following source data is available for figure 6:

**Source data 1.** Maximum length and maximum width of marine snails from Boi Yai Island and Lobi Bay.

DOI: https://doi.org/10.7554/eLife.46961.009

higher degree of use wear intensity. This pattern was recorded across oyster and *Thais bitubercularis* tools. The differences in oyster tools are most likely influenced by the oyster size difference reported between the locations. The intensity of use wear seen on oyster tools of Boi Yai Island, however, is unlikely to have developed during a single foraging event (*Haslam et al., 2016c*). *Thais bitubercularis* tools used on Boi Yai were characterised by the development of substantial central pits,

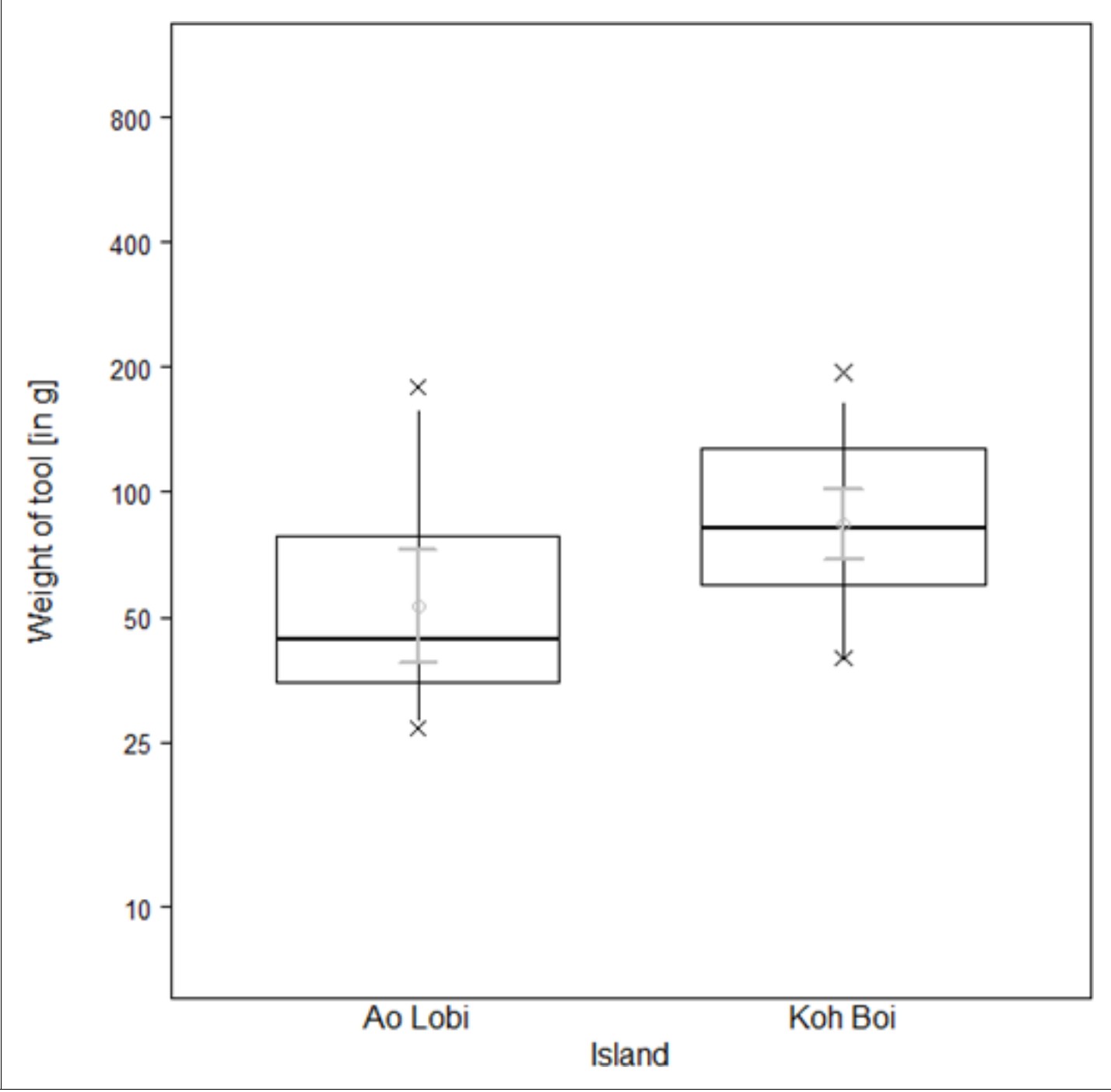

**Figure 7.** Tools selected by wild macaques to crack open oysters at Lobi Bay and Boi Yai Island. The plot is showing all quantiles and the CIs (grey). (For underlying data, see *Source data 1*).

DOI: https://doi.org/10.7554/eLife.46961.010

suggesting multiple re-use events, whilst the hammerstones used to process the same species on Lobi Bay showed few signs of percussive damage. For marine snails (*Nerita* spp., *Monodonta Labio* and *Morulla* spp.), there was no significant difference in use wear intensity. It is reasonable to suggest that the relative fragility of these prey species means that little force was required to open them. Consequently, use wear intensity is low for these tools on both islands.

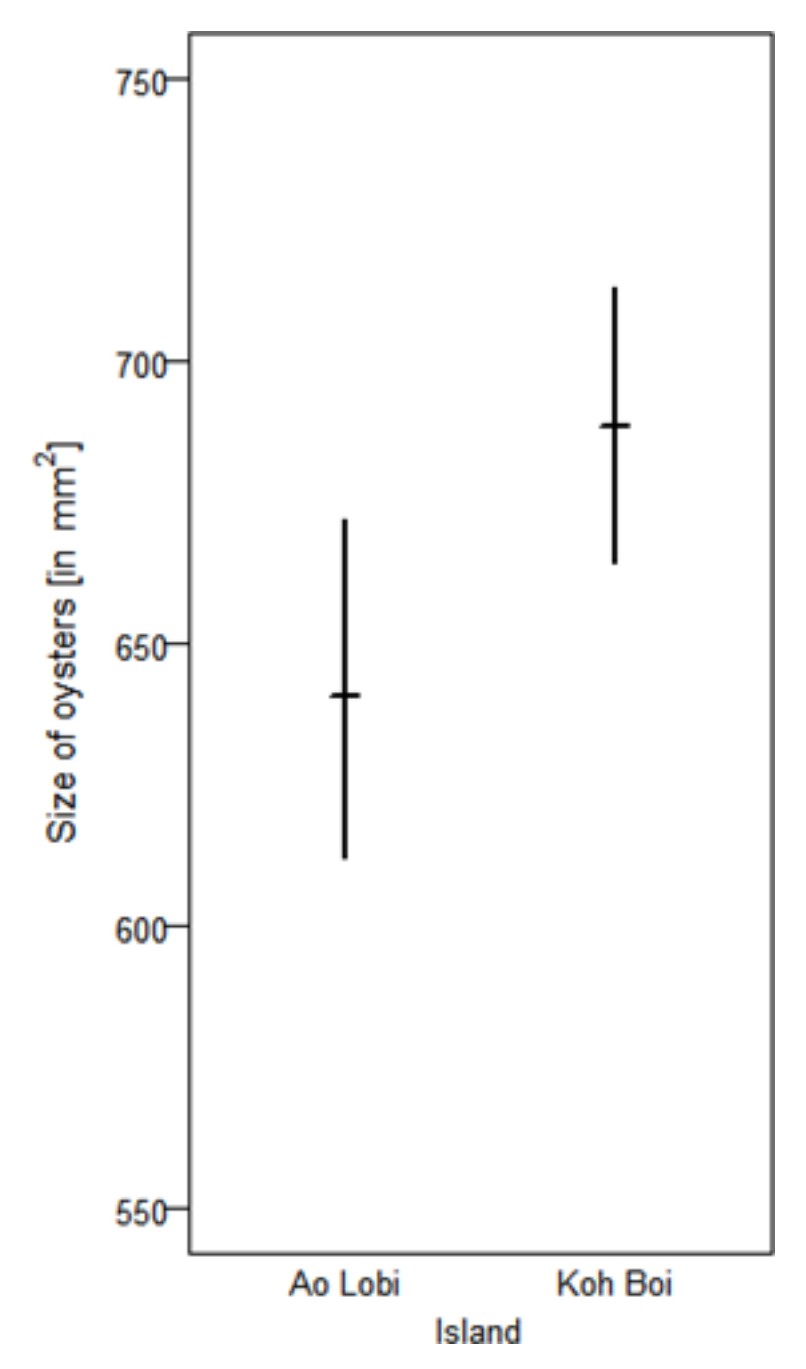

**Figure 8.** Size of oysters available at Lobi Bay and Boi Yai Island. The plot is showing the mean with bootstrapped 95% confidence intervals.

DOI: https://doi.org/10.7554/eLife.46961.011

The following source data is available for figure 8:

**Source data 1.** Maximum length and maximum width of oysters on Boi Yai Island and Lobi Bay.

DOI: https://doi.org/10.7554/eLife.46961.012

The lack of evidence of re-used hammerstones on Lobi Bay could be influenced by the tidal forces that potentially remove used tools from their original location. Lobi Bay however, appears relatively sheltered compared to Boi Yai Island as it lies within a large bay. Future research will be able to document the effect of tidal forces on tool movement and will clarify the availability of used tools

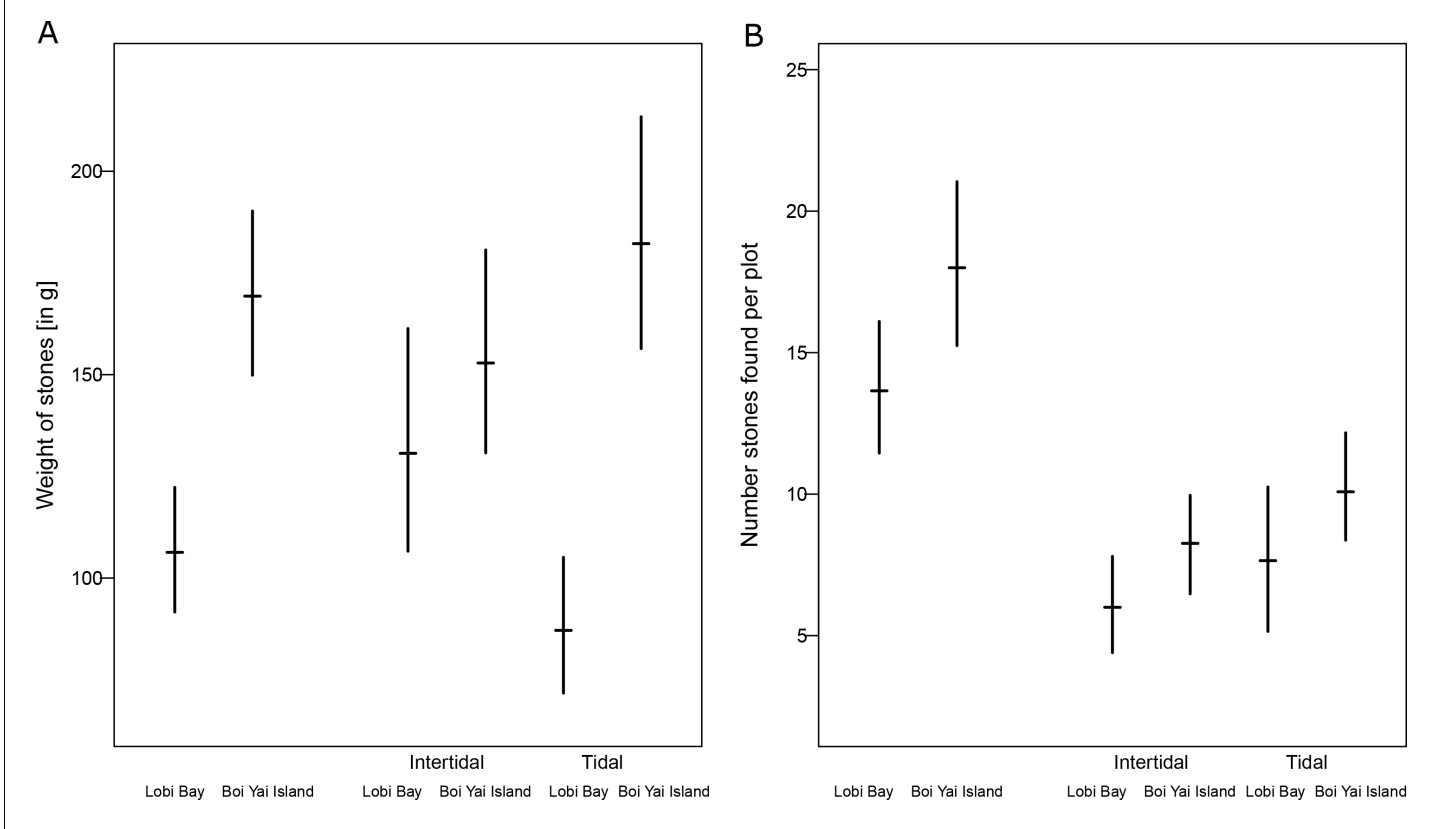

**Figure 9.** Availability of raw material at Lobi Bay and Boi Yai Island. (**A**) Weight of stones available at point transects. (**B**) Number of availability of stones. The plots show the mean with bootstrapped 95% confidence intervals.

DOI: https://doi.org/10.7554/eLife.46961.013

The following source data is available for figure 9:

**Source data 1.** Measurements and weight of stones available in the tidal and intertidal zone on Boi Yai Island and Lobi Bay.

DOI: https://doi.org/10.7554/eLife.46961.014

between populations. A single example of a granite hammerstone was identified within the Boi Yai assemblage (*Table 1*). This igneous rock is more dense than the available limestone rocks available to the macaques which may have been a factor in its selection and re-use as a hammerstone. Of course, the re-use of this rock may be coincidental, and it should be remembered that any available granite has been geologically transported to Boi Yai Island from elsewhere and is therefore very rare (*Luncz et al., 2017a*).

As shown previously, raw material availability is consistent between both islands, with limestone constituting most of the available raw material. Scarcity of raw material, cannot, therefore, explain the apparent preference for tool re-use on Boi Yai Island. Indeed, raw material availability on Boi Yai Island is greater than that found at Lobi Bay. This would suggest that macaques on Boi Yai Island, unlike their neighbours, preferentially re-use tools rather than choose from readily available and suitable local stones.

The tool use variation observed between these two island primate populations meets the criteria for tool curation as discussed in the hominin archaeological literature. *Shott and Sillitoe (2005)* define lithic tool curation as the 'ratio of realized to maximum utility' of a tool. The varying degrees of repeated use, shown by use wear assessments of the hammerstones between the two islands, coupled with the similarities in primate species, raw material availability, processed prey species, and local ecological settings, suggest that the macaques on Boi Yai Island are successfully extracting a greater degree of utility from their hammerstones compared to those used on Lobi Bay. These

variations in the intensity of tool use suggest a primate example of curated stone tool use on Boi Yai Island compared to those on Lobi Bay.

Previously, tool differences between wild macaque groups were only known to be influenced by prey size (*Tan et al., 2015*; *Luncz et al., 2017b*; *Gumert and Malaivijitnond, 2013*) or sex of the tool user (*Gumert et al., 2011*). Furthermore, it has been shown that the frequency of tool use in long-tailed macaques might be influenced by genetics, where the Burmese subspecies (*Macaca fascicularis aurea*) and hybrids expressing a more Burmese phenotype use tools in higher frequencies (*Gumert et al., 2019*). Our research sites in the Ao Phang Nga National Park lie within a natural hybrid zone and phylogenetic differences between groups cannot yet be fully excluded. Preliminary data show similar degrees of hybridization between groups (*Malaivijitnond et al., 2007*). The demographics (assessed from video footage, *Luncz et al., 2017a*) do not differ between the two populations. Males and females of all age classes use stone tools, it therefore seems unlikely that the difference in stone tools between the two islands stem from these factors. Muscle strength, individual proficiency and strike accuracy may all be contributing factors to the development of use wear, as these tools were most likely repeatedly used by multiple individuals over numerous tool use episodes. However, similar demographics between both macaque populations make it unlikely that these factors contributed to the differences found in use wear intensity. Additionally, by combining our new results here with our previous findings of tool use by macaques in the Gulf of Thailand (*Luncz et al., 2017b*), we were able to identify that macaques of Lobi Bay and Boi Yai Island do not forage on *Cerith bifasciatus*, despite its widespread abundance. Macaques in the Gulf of Thailand, however, regularly exploit this species.

These results partly support previous research that has shown that tool selection can be explained by ecological, biological and environmental factors (*Gumert et al., 2011*; *Gumert and Malaivijitnond, 2013*; *Luncz et al., 2017b*). However, the differences in use wear intensity and therefore frequency of re-use of hammerstones on Boi Yai Island remains unexplained by these factors.

Our new results show differences in use wear intensity and the extent to which macaques unintentionally modify the shape of their tools through use. This would suggest that tool use across a synchronous landscape need not be uniform, and highlights the potential for the development of synchronic tool use variation within species. Indeed, tool variation may be conditioned by cultural factors as well as responses to ecological conditions. Tool use has been shown to be influenced by social learning in several species (*van Schaik et al., 1999*; *Whiten, 2000*). To date, distinct cultural differences in technological behaviour in living primates has mainly been associated with the ape clade (*Whiten et al., 1999*; *Whiten and Boesch, 2001*; *van Shaik, 2004*; *Luncz et al., 2012*).

As this study, however, is dependent on primate archaeological data, as opposed to direct behavioural observations, it is important to note that its interpretations are subject to the same or similar constraints as found in the early hominin archaeological record. Identifying underlying reasons for behavioural variation in the animal kingdom often requires direct observations of differential tool use despite similar ecological situations between populations. Direct observation of behaviour is, however, impossible when attempting to assess cultural variation in the early archaeological record, and as such a cultural driver for observed lithic variation should only be invoked once non-cultural factors have been ruled out. Non-cultural factors may include, raw material availability and quality, ecological variation, adaptation to climatic change and genetic variation. In the absence of direct observations primate archaeology must also strive for similar goals in identifying possible cultural variation.

The observed artefactual variation between the two island macaque populations described here, although offering an opportunity for investigating the reasons for variation in tool using macaque populations, also provides a unique primate analogy to discuss possible drivers behind early lithic variability in the hominin archaeological record from a similar perspective as that available to archaeologists. Thus, it challenges us to consider the variables in the archaeological record that might drive assemblage composition but cannot be immediately tied to ecological or raw material constraints. Within the field of palaeolithic archaeology, understanding the cultural factors behind lithic variation has been at the forefront of research and has led to two primary theoretical frameworks, the cultural historic approach, which stressed the importance of typological tool categories to develop sequences of hominin cultural traditions (*Leakey, 1971*; *Isaac, 1977a*) and the chaine-operatoire approach (*Roche and Texier, 1996*; *Roche et al., 1999*; *Delagnes and Roche, 2005*; *de la Torre and Mora, 2005*; *de la Torre, 2011*), which has been used to reconstruct the technical processes and the social

acts involved in tool use. At its core, however, is the working assumption that hominin lithic variation is a product of cultural differences both between geographically and temporally distinct groups as well as between hominin species (*Sellet, 1993*).

The other side of the theoretical divide is an ecological approach to lithic variation. After identifying limitations in the cultural historic approach for understanding the wider effects that the environment and ecology had on hominin behaviour and stone tools a palaeoecological approach has been put forward (*Isaac, 1977b*; *Toth, 1982*; *Toth, 1985*; *Braun and Harris, 2003*; *Braun et al., 2010*). This framework stressed the importance of viewing hominin stone tools as being intrinsically linked to external environmental and ecological factors. These included, ecological factors (*Braun and Harris, 2003*; *Braun et al., 2010*), raw material availability (*Stout et al., 2005*; *Harmand, 2009*; *Braun et al., 2008b*; *Braun et al., 2009a*; *Braun et al., 2009b*; *Goldman-Neuman and Hovers, 2012*) and functional factors (*Toth, 1985*).

One of the strengths of primate archaeological studies is the ability to test long standing fundamental archaeological theories and assumptions by applying them to stone tool using living primates (*Haslam et al., 2009*; *Luncz et al., 2016b*). Our findings suggest that when raw material availability, species differences, functional differences and environmental and ecological conditions are shared and do not explain stone tool variation the effect of cultural differences as a driving factor behind lithic technological variation in hominins must not be overlooked (*Koops et al., 2015*; *Luncz et al., 2015*; *Pascual-Garrido, 2019*). Our results highlight the possibility that tool selection in old world monkeys might also be affected through social learning and therefore might classify as a cultural behaviour.

This study builds a platform for long-term research focused on the group specific behaviours in populations of long-tailed macaques through the archaeological analysis of tool. Further archaeological research at the site coupled with direct observations will reveal the stepwise development of use wear patterns in long-tailed macaques. This highlights the potential of primate archaeology for answering questions regarding the ecological and socio-cultural factors which affected the development of stone tool use in our own lineage, given the absence of direct observations on early hominins (*Westergaard, 1998*; *Wynn et al., 2011*).

Unfortunately, the number of islands where long-tailed macaques are known to use tools are few and, in each instance, the cultural behaviour is possibly unique. The Andaman Sea coast of Thailand is renowned for its natural beauty and attracts increasing numbers of tourists. Increased human contact is known to affect stone tool behaviour by altering natural foraging behaviours (*Gumert et al., 2013*; *Luncz et al., 2017a*). Long-tailed macaques are experiencing profound population threats (*Eudey, 2008*). However, it is clear that these potentially rare, and certainly distinctive, examples of tool behaviour in old world monkeys need to be preserved and protected from human interference. Otherwise examples of extraordinary material culture may be lost before they are even discovered.

## Materials and methods

### Study sites

We collected data in January and February 2017 and 2018 on two shorelines at islands in the Ao Phang Nga National Park. The first site, Lobi Bay, is located in the northwest tip of Yao Noi Island (8° 10′ 51.06′ N, 98° 37′ 42.52′ E, *Figure 1A*). The second site is 9 km to the southwest, on the southern end of Boi Yai Island (8° 07′41.09′ N, 98° 33′ 25.04′ E, *Figure 1B*). Below the shoreline is an intertidal zone extending up to 200 m at spring tides. The area above the shoreline is comprised of steep coastal forest. Because the macaque populations at Boi Yai Island and Lobi Bay are not habituated to the presence of humans, the field of primate archaeology offers the most appropriate way of reconstructing tool selection and analyse damage pattern (use wear) (*Haslam et al., 2013*; *Haslam et al., 2016a*). This approach however also imposes limitations, as we are not able to differentiate between the sex and age of the tool user. However, tool use is a social activity in long-tailed macaques, with most of the group foraging together along the shore. Previous studies at Lobi Bay and Boi Yai Island using camera traps to document nut cracking behaviour confirmed that both groups comprised mixed demographics, where males and females in all age ranges use tools habitually (*Luncz et al., 2017a*). Furthermore, during low tide, we were able to observe shellfish foraging of monkeys on the shore from far away. Identification of individual group members was not possible,

however since shellfish foraging is a social even in long-tailed macaques, most group members were counted simultaneously foraging together. Unfortunately, camera traps are not feasible in the ocean setting of shellfish foraging and therefore our study relies on indirect evidence of stone tools and prey remains. Based on the mtDNA phylogenetic analysis, these two populations are genetically similar (Malaivijitnond S., in prep.).

At both sites, Boi Yai Island and Lobi Bay, wild macaques use stone tools to exploit rock oysters (*Saccostrea cucullate*) and a variety of marine snails including *Nerita* spp., *Monodonta labio*, *Morula* spp., and the larger *Thais bitubercularis*. Due to the similar morphological nature and size of *Nerita* spp., *Morula* spp. and *Monodonta labio*, we combined those species to analyse use wear patterns and they are hereafter referred to as marine snails. As macaques have been reported to frequently consume *Cerith bifasciatus* marine snails, we also collected information on their availability.

Boi Yai Island and Lobi Bay on Yao Noi Island are limestone outcrops within the Ao Phang Nga National Park which is underlain mainly with Permian carbonate rocks which were deposited about 295–250 million years ago (*Kuttikul, 2015*). The geological profile of available raw material across the shoreline is predominantly limestone, with the occasional rock formed of igneous rocks such as granite cobbles, which do not occur naturally on the Islands but may have been transported to the islands by wave action. There are occasional granite outcrops in other areas within Phang Nga bay (*Charusiri et al., 1993*), and this is a tsunami prone region (*Kongapai et al., 2016*) which could transport stone material to the research sites.

At both sites, Lobi Bay and Boi Yai Island, we conducted line transects along the shore and collected all stone tools previously used by the macaques to forage on marine prey. Stones were identified as tools if they showed fresh percussive marks and were found within 20 cm of identifiable prey remains. This distance was determined as the maximum to reasonably associate prey with tool. For each tool we identified which prey it had been utilised for. Where there was any doubt regarding the association of a tool with a particular prey, the tool was excluded from this study.

All tools were subsequently cleaned, weighed and all raw material types were documented. Use wear was recorded following a modified technique adopted by *Haslam et al. (2013)*. Each tool was divided into ten zones and use wear (pitting, crushing, and fracture) was recorded for each zone. The 3-level system adopted by *Haslam et al. (2013)* to grade severity of use wear did not provide the necessary resolution to distinguish variations as it originally was designed to assess functional

**Table 2.** Grading of use wear by zone (adapted from *Haslam et al., 2013*)

| Use Wear Grade (numerical) | Use Wear Grade (descriptive) | Pitting | Crushing | Fracture |
|---|---|---|---|---|
| Definition | | Distinct indentations from discrete strikes that can culminate in the creation of more general craters in the stone surface | Rounding and flattening of the tool surface, especially the protruding parts on the points and edges. | A breakage of the tool caused by chipping or flaking |
| Grade 0 | None | No trace | No trace | No trace |
| Grade 1 | Slight | Trace - minimal and isolated damage. Isolated points of impact | Trace - minimal and isolated damage | Trace - minimal and isolated fracture |
| Grade 2 | Medium | Overlapping impact points that have formed a coherent platform of damage | Surface clearly damaged but limited evidence of repeated use. | Moderate - larger fracture affecting < 30% of the use zone |
| Grade 3 | High | Cumulative damage with a pitting to a depth of 2 mm | Cumulative - rounding can be observed and felt. | Extensive - Fracture affecting between 30% and 60% of the use zone |
| Grade 4 | Very-High | Cratered - a larger more extensive indentation that is more than 2 mm deep and 5 mm in diameter. | More progressive rounding that has produced a flatter surface on the use zone. | General - fracture extending beyond 60% of the use zone |

DOI: https://doi.org/10.7554/eLife.46961.015

use. Therefore, for each type of use wear, we adopted a use wear grading system (UWG) describing five different use wear intensities (*Table 2*).

The system designed by Haslam categorised use wear into the following three subsets: (1) Trace - superficial and isolated points of damage only, possibly incidental; (2) moderate – the stone surface was clearly damaged but this damage was spatially restricted with only minor evidence of repeated wear; and (3) extensive - cumulative damage that may cover a significant portion of the use-zone. The aim of Haslam et al was to use the wear patterns on the tools to identify foraging behaviours and therefore a limited scale of differentiation of damage intensity was sufficient. Here the objective was to specifically compare use wear intensity and potential re-use of tools used by two discrete populations of macaques. Therefore, a three grade scale was insufficient as it can skew results in the direction of amplifying observed differences. The emphasis was placed on designing a key that was reliable in differentiating between the higher end of the use wear scale to more accurately interpret the likelihood of reuse. Using the UWG definitions (*Table 2*), multiple separate analyses of the material was undertaken by two researchers to ensure replicability of results.

In respect of *pitting*, where use wear was rated as grade 4, we recorded the length, width and depth of each pit. Pit depth was measured digitally using a pit gage across an axis that lay across two opposing edges of the pit and crossed the deepest part of it. This method which anchors the arms of the gage at the pit edge and lays across the deepest part ensures that results are replicable. Given that the UWG distinguishes between grades 3 and 4 (*Table 2*) only based on a pit being greater than 2 mm deep, digital microscopic measurement was not considered necessary.

To compare use wear intensity, we identified the highest score of damage inflicted by the macaques on each tool and compared that between sites. To assess tool selection patterns between sites we compared the weight of each used tool in regard to the locally available raw material. To analyze whether the weights of stone tools used to crack open oysters or marine snails differed between the two observed populations we ran two linear models (LM), one for each prey species, which each included the population (Boi Yai Island and Lobi Bay) as fixed effect. As response we used the weight of the stone tools used to open oysters or snails. Prior to running the models, the response variable was log transformed to achieve a more symmetrical distribution. For both models we checked whether the assumptions of normally distributed and homogeneous residuals were fulfilled by visually inspecting a plot and the residuals plotted against fitted values. In both models we found no obvious deviations from these assumptions. We additionally tested model diagnostics using the R functions ''dffits,'' ''dfbeta'' and ''cooks.distance''; and checked for leverage and found that no assumptions were violated.

Furthermore, we compared the intensity of the tool damage between the sites by comparing the occurrence of the different use wear intensities. The damage with the highest intensity rate was used in a Mann-Whitney U-test separated by prey species.

## Environmental comparisons

Because the macaques are not habituated, direct observations during shellfish foraging along the shore are not possible. In order to collect used tools, we inspected the areas at the shore where monkeys were directly observed foraging from afar during the low tide and repeatedly returned after low tide to assess the area for more tool use. On Boi Yai Island this totalled a shoreline of approximately 1100 metres, whilst on Lobi Bay this resulted in a shoreline length of approximately 900 meters.

In order to assess the availability of stones for each population, we used point transects (plots) along line transects across the intertidal and tidal zone. The tidal zone describes the area between high and low water and the intertidal zone is located above the tidal zone. The intertidal zone floods only during the peaks of the tide cycles and remains dry for long stretches between high tides. It is inhabited by hardy sea life, such as barnacles, marine snails. Because the intertidal zone is accessible for much greater periods than the more frequently flooded tidal zone macaques would have greater opportunities in a given timespan to access raw material and prey. For this reason, we compared the stone availability separated within the intertidal and tidal zone at a given longitudinal positions.

Each plot was randomly selected 100 meters apart along the line transects. This resulted in 20 plots in each zone in Lobi Bay and 24 plot on Boi Yai Island. Within each plot all stones larger than 2 cm in maximum length were weighed (stones smaller than this were considered too small for macaques to utilise as tools). Due to the large amount of available stones each transect plot was limited

to 20 by 20 cm. Plots were arranged in pairs where one plot was located in the tidal zone and one in the intertidal zone at the same longitudinal position on the transects. Tidal and intertidal plots were at least 20 meters apart from one another. Separated for island (Boi Yai Island and Lobi Bay) and location (tidal and intertidal) we bootstrapped i) the number, ii) the weight of the stones we found per transect 1000 times and compared the confidence intervals at the level of 95% between the islands. Additionally, we applied two sample t-tests.

We assessed the availability and size of prey species at both sites within 1 $m^2$ plots along the same line transects as the stone plots, spread 100 meters apart from each other. This resulted in 10 plots in Lobi Bay and 12 plots on Boi Yai Island. Within each plot the frequency, length and width of available prey species were documented. To increase the sample for the size measurements we additionally searched for marine prey along the shore at both sites.

To compare the size of oysters, we measured the maximum length and maximum width and calculated a volume for 341 oysters in Lobi Bay and 436 oysters on Boi Yai Island. We bootstrapped the volume for each site 1000 times and compared the confidence intervals at the level of 95% to each other. Additionally, we applied a two-sample t-test. To test for differences in snail size between the sites, we conducted the same procedure for four snail species (*Cerith bifasciatus* ($N_{Lobi}$ = 21, $N_{Boi}$ = 73); *Nerita* spp. ($N_{Lobi}$ = 120, $N_{Boi}$ = 119); *Thais bitubercularis* ($N_{Lobi}$ = 14, $N_{Boi}$ = 88)).

To investigate the availability of marine snails we bootstrapped the number of snails we found at each site and on each transect 1000 times and compared the confidence intervals at the level of 95% between the sites. Additionally, we applied two sample t-tests. As oysters are abundant throughout the shoreline and did not pose a limitation of availably, they were excluded from the comparisons of prey availability.

All models, bootstraps and tests were implemented in R version 3.5. (*R Development Core Team, 2015*). The LM was fitted using the function lm, and the Mann-Whitney U-test with the function wilcox.test.

## Acknowledgements

The National Research Council of Thailand permitted LVL and MS to conduct research in Thailand, and the Department of National Parks, Wildlife and Plant Conservation gave permission to enter and conduct research in Ao Phang Nga National Park. We thank the National Park rangers of Boi Yai Island for their logistical support. We thank the editor, reviewers Amanda Tan and Jessica C Thompson, two additional anonymous reviewers and Alexander Mielke for helpful comments on this manuscript. This research was funded by the Leverhulme Trust and by the German Primate Centre in Goettingen, Germany. During writing TP was supported by the British Academy Postdoctoral Fellowship (pf170157).

## Additional information

### Funding

| Funder | Author |
| --- | --- |
| Leverhulme Trust | Lydia V Luncz |
| Primate Center Göttingen | Lydia V Luncz |

The funders had no role in study design, data collection and interpretation, or the decision to submit the work for publication.

### Author contributions

Lydia V Luncz, Conceptualization, Data curation, Formal analysis, Funding acquisition, Investigation, Methodology, Writing—original draft, Project administration, Writing—review and editing; Mike Gill, Data curation, Writing—original draft; Tomos Proffitt, Investigation, Visualization, Methodology, Writing—original draft, Writing—review and editing; Magdalena S Svensson, Data curation, Writing—review and editing; Lars Kulik, Investigation, Methodology, Writing—original draft,

Writing—review and editing; Suchinda Malaivijitnond, Conceptualization, Project administration, Writing—review and editing

### Author ORCIDs
Lydia V Luncz https://orcid.org/0000-0003-2972-4742

### Decision letter and Author response
Decision letter https://doi.org/10.7554/eLife.46961.020
Author response https://doi.org/10.7554/eLife.46961.021

## Additional files

### Supplementary files
• Transparent reporting form DOI: https://doi.org/10.7554/eLife.46961.016
• Source data 1. Measurements and use wear intensity scores of tools collected on Boi Yai Island and Lobi Bay. Source data for *Figures 4*, *5* and *7.*
DOI: https://doi.org/10.7554/eLife.46961.017

### Data availability
All data generated or analysed during this study are included in the manuscript and supporting files.

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
