## [Decision Letter]

Thank you for submitting your article "Group specific stone technology suggests cultural diversity in Old-World Monkeys" for consideration by *eLife*. Your article has been reviewed by four peer reviewers, including Jessica C Thompson as the Reviewing Editor and Reviewer #2, and the evaluation has been overseen by Detlef Weigel as the Senior Editor. The following individual involved in review of your submission has agreed to reveal their identity: Amanda Tan (Reviewer #4).

The reviewers have discussed the reviews with one another and the Reviewing Editor has drafted this decision to help you prepare a revised submission.

Summary

This manuscript represents a detailed and well-written investigation into the archaeological signature of macaque tool use on two neighboring populations in Thailand. It is impressive in its scope and detail. The factors investigated here have direct relevance for tool use across the primate record and provide tantalizing details of how we might use the meticulous documentation of this behavior amongst living primates to understand aspects of early hominin behavior that are not directly observable.

The authors do a good job of working their way through layers of logic to demonstrate that species/population differences, raw material availability, and resource package size (in short, ecological/environmental and raw material constraints) do not explain all the differences that they observe in tool use. Thus, it challenges us to consider the variables in the archaeological record that might drive assemblage composition, but can't be immediately tied to ecological or raw material constraints. The authors also make an excellent point about conservation, as this behavior in wild primates may be threatened by increasing human interactions with them. For these reasons, it is a worthy contribution. However, it needs revisions before advancing.

From their results, the authors conclude that cultural differences – transmission of distinct and population-specific behaviors through social learning – is responsible for the patterns that cannot be explained by other variables. This is the same logic used by archaeologists when documenting traces on objects or their associations with one another. The difference is that it is impossible to observe the behavior that led to these traces and associations in the archaeological record, but it is possible to observe this amongst living organisms. Since direct observation was not possible with the macaques, this is essentially an archaeological study carried out in the present day.

The reviewers concur that the strength of this paper lies in demonstrating how arguments that are being used in archaeology can have a resonance in living animals, as well as application to the archaeological record. This paper therefore could represent a significant advance in the joining of the fields of archaeology and animal behavior, but this would require a reframe of the paper to emphasize the verifiable aspect of the paper, which is the use of archaeological methods to suggest culture, rather than on demonstrating culture itself is the cause of the tool differences. The work would do more to advance the field by contemplating a wider range of possible mechanisms for variation in tools instead of sticking with the current dichotomy of culture (generally considered exciting) or not culture (generally considered boring).

The reviewers also concurred that in the absence of direct observational data, the paper cannot be convincingly framed in terms of primate "culture" (although this may be true, it does not meet the observational tests required in animal behavior research). We cannot be even sure that effects found for the marine snail are not the result of the activity of a single individual (or a matriline), who would favor the use/re-use of certain stones. Later in the manuscript, the reviewers mention that previous studies have documented that make and female macaques will use different sized tools. Could this account for some of the variation? As Bandini and Tennie, 2017 have suggested, it is particularly important to show that any example in the wild is not the result of re-invention by individuals, which would make the argument weaker for all animal cultures in general.

Again, all reviewers saw much value in the fact that macaques faced with similar ecological conditions and similar requirements, with similar genetics, do not produce identical archaeological records. Whether this is attributable to culture or not at this point becomes less important than the fact that this is exactly what we must deal with in the archaeological record. This manuscript suggests that investigations of primate tool use may provide us with insights into the mechanisms of social learning. This would be particularly interesting because of the importance that evolutionary psychologists place on certain types of social learning (imitation, etc.). However, again the social context of the tool use and learning is not directly observable here, and the authors do not have the social data that would tell us exactly how this information is transferred between individuals. Therefore, they should remove sentences about social learning, because they infer, rather than observe, that this happened.

In the end, all reviewers agreed that the authors should reframe this manuscript to emphasize its strengths relative to the archaeological record and back away from claims of culture and social learning as the primary focus. In light of this reframing, the manuscript mainly requires changes to the background and discussion. "Culture" or social transmission processes more broadly are undoubtedly involved in macaque tool use, but these can be discussed more thoughtfully with regards to how social learning might influence behavior and artifact production. Proposing these possible explanations would direct further research, something that is actually possible with the macaque study system, unlike with the hominin archaeological record. Therefore, the authors should use the opportunity to report observed differences in material objects and environments, and set up hypotheses about what different stone traces COULD be reflecting, much as archaeologists do. This could also involve suggestions for future work that actually uses archaeological methods. For example, if some are multi-use tools, which is in and of itself very interesting, it is especially important to be able to document all the resources on which a single tool was used and not just the one that was last used and discarded nearby. In future, and in the absence of observational data, the authors might consider taking samples from the rock surfaces for proteomic or residue analysis, much as archaeologists do.

The reviews been consolidated below into the major points that should take place to make this manuscript acceptable for publication in *eLife*. Because many of the points include useful discussion and suggestions for additional references, much of the original wording has been retained, along with the helpful suggestions therein. This makes for a somewhat long decision letter that can be boiled down to nine major points.

Essential revisions

Below is a summary of the key changes that should take place to make this manuscript acceptable for publication in *eLife*:

1) Clear integration of this research with the previous study. As a Research Advance, a more substantial review of the results of the previous study would be beneficial. How and why are these studies different and how are they alike? It is not clear enough to readers who are unfamiliar with macaque tool use sites that the two studies were conducted with two entirely separate populations at different sites. Also, since the original study found that ecological parameters do appear to be driving some of the variation in tool use, the deviation from this in the current study becomes even more interesting. The previous study capitalized upon the size differential between the two macaque populations to examine whether habitual tool-aided exploitation by macaques could lead to reduction in size and abundance of shellfish prey. This contributed to the long-standing debate in coastal archaeology as to whether the reduction in shellfish sizes in the Middle Stone Age could have been due to human exploitation of marine resources. Here, this study is showing that primate tool artifacts also vary between populations. The authors should emphasize these potential contributions via primate archaeology, and how further behavioral work could help with interpreting the lithic record (particularly with respect to the concerns raised above).

2) Re-framing the significance of the work. This should be with respect to what is directly observable, and more specifically address its implications for the archaeological record of early hominins, rather than simply labelling patterns as cultural or non-cultural.

3) Clarifying possible alternative explanations than "culture". The lack of information about the sex and age of individuals does not invalidate this study, and in fact makes it more akin to archaeological work. However, all possible sources of variation should be mentioned as possible explanations. One possibility is providing information on the number of individuals or different age on the two islands. If the islands do not differ in their age and sex demographics then it would seem unlikely that this is a reason for these differences. However, the authors should still discuss other possibilities. How confidently can we disentangle the effects of use frequency from users' strength/proficiency/strike accuracy on wear patterns? Macaques do differ in proficiency, and some tool users will be able to crack food items with fewer strikes or might be more accurate with their strikes. These users would conceivably create fewer and less diffuse use wear than users who require many strikes to crack food items. The latter might create use wear that would be scored as more intense or extensive but interpreting this as re-use would be erroneous. Perhaps differences in use wear patterns between both populations could indicate differences in population level proficiency? In spite of the authors' claim that the snails are morphologically similar, there could be subtle ecological differences that could explain a different use of stones by the monkeys on the two islands. For instance, there might be a higher proportion of a certain type of snail in one of the two islands, which would make the re-use of certain stones more efficient. There is also a cognitive argument that can be made about tool selection, e.g. functional fixedness. Macaques on Boi Yai have to use the heavier tools to crack oysters because they are bigger than on the other island. That means the heavier tools can be functionally fixed to be used with oysters. Macaques then use lighter rocks (the ones that are not functionally fixed) to crack the snails. On the other hand, on Lobi Bay, there is no functional fixedness at work, and there is an additional problem: stones appear to be rather lighter in general, in comparison to the ones in Boi Yai, at least according to Figure 8: it is unclear is a rock of less than 100g is enough to break a snail, so macaques could be drawn there to take heavier stones. On that note, there is confusion about the difference in weight that can be found in the figures. In Figure 8, weight ranges from less than 100 to around 200g. On Figure 5, weights are rather around 400-800g. If these are estimates obtained from the model, the authors should say it.

4) More in-depth treatment of the archaeological and primate archaeological literature. The mention of pounding tools as part of the toolkit of the LCA could likely include references to Rolian and Carvalho, 2017. The authors should also reference John Shea's recent paper (Evol Anthropology, 2018) about tool use as habitual versus occasional, and perhaps Thompson's (2019 Current Anthropology) piece on pounding tool use as the precursor to flaked stone tool use. Elsewhere in the Introduction the authors mention the idea of the extraordinary advantages that tool use provides those individuals that implement these strategies. While this makes sense to me theoretically, there are only a few examples where this has been explicitly tested. One example would be Morgan et al., 2016 who explicitly tested the significance of tools. Similarly, the final sentence of paragraph four on the Introduction that mentions that fact that primate tool use leaves an archaeological record could benefit from references to Carvalho et al., 2012. This would however call for a review of other papers that have been published on the topic, and what this particular paper brings in addition.

5) Clarification and de-emphasis of multiple use tools (or removal of this argument). The argument on multitool or curated (in the sense used by the authors here, but see point 6 below) usage is, right now, too weak. The idea of multi-use tools is interesting, but cannot be used here as one of the main arguments for differences between groups. How do we know it was used on multiple prey-species? The authors state that if a specimen could not be identified as being associated with a single species then it was not included in the study, so this would contradict that. Furthermore, there are only 7 stones observed, and the researchers haven't been able to observe that they were really used for different purposes. In general, the discussion of the Boi Yai island tool is problematic (L298+). Following the argument of the authors, there is not much use wear for snails because little force is needed. But then, the same argument should apply for oysters: because they are bigger, then more force should apply and thus likelihood of use wear increases. And once again, functional fixedness or conservatism, which are both mechanisms acting at the individual (rather than 'cultural') level, can kick in, and explain the re-use, because it is unclear what macaques consider the same substrate and what is not. In fact, it is unclear whether re-use is evidence of culture in any animal species so far (see for example Hobaiter et al., 2014, Plos Biology), rightly so because of the possibility of individual characteristics (it is much more parsimonious than to claim norms for instance). In addition, if some tools were indeed reused, how can the authors be sure that the findings are not contributed by single or several individuals? The authors do not have behavioral data from this population to confirm this, but observations of tool use by macaques at other sites should make them aware that some individuals more than others have stronger tool preference and retain tools for longer periods of time than others. This possibility must at least be discussed. The authors interpret scarring on different tool surfaces as a repurposing of the tool for different food items. This is based on previous findings on one population of macaques on Piak Nam Yai island that tool points were predominantly used on oysters while faces were used on gastropods (Haslam et al., 2013, Tan et al., 2015). More current research on yet another population on Koram Island however, is finding that the macaques there predominantly use tool points regardless of whether or not they were processing sessile oysters or motile gastropods. This work is preparation and not yet published, so it is understandable that the authors not consider this in their interpretations of use wear at present, but it does signal a need for greater caution. A final possibility might be that the difference in use intensity between two populations is related to differences in tidal conditions. From Figure 1, it appears that the Boi Island site is relatively more sheltered from the open sea compared to Lobi Bay. We know tool-aided coastal foraging can take place only during the low tides, and used tools left along the shore can be washed away by wave action. Is it possible that the macaques on Boi Island have greater access to the coast, and that tools on Boi Island are less likely to be washed away by wave action?

6) Clarification of terminology to connect archaeology and animal behavior literatures. The authors use the word "curation" several times in the manuscript. This is an interesting use for the word in this context, but the authors need to be more explicit about what they are describing. The word curation in relation to tool use stems from Lewis Binford's investigation into the variance in resource and land use by modern hunter-gatherers and his gatherer vs. forager dichotomy (Binfrod, 1973; 1979; 1980). However, largely stemming from the vague nature of this description, Shott and others have reviewed this concept. The most current concept of curation in archaeological contexts is the realized utility of a tool relative to the actual utility of a tool. This is after an extensive review of what is meant by curation by Shott (1996 as well as subsequent reviews of use life Shott and Sillitoe, 2006). It would be exciting to see primate archaeology begin to use the vocabulary of Paleolithic archaeologists in this way, but if the authors intend to do so they should describe what they mean by curation and implement it into this manuscript (with references to the original literature). If they are not going to do that, then they should use a different vocabulary altogether. For example, the term "repeated curation" is not how archaeologists would use this term, and curation would not be something that is repeated. This would conflate curation with re-use. If they use the term in the way that Shott describes it they could quite easily accommodate the idea that the Boi Ya macaques are extracting a greater amount of utility from the tools then those at Lobi Bay. In archaeological literature this would be described as differing levels of use life and site use intensity usually associated with ecological differences. However, I think the authors have convincingly shown that this is unlikely, especially since the greater levels of intensity of use are not seen in all species. Another example of terminology that does not connect the two bodies of literature is in how the authors describe transects as "background material" which is used to describe the locally available stones that were not selected for tool use. It is excellent that they are providing the data for this, which is directly comparable to archaeological data which do the same thing. However, in that case it should be referred to as "locally available raw material" as opposed to "background material" because it will make conversations between archaeologists and primate archaeologists more productive to be using the same vocabulary.

7) Description of raw material. The description of the rock types used by the macaques indicates that they use mostly limestone but also use calcite (which seems like a very bad option for cracking shells but I guess it is what you need to use if you have no other options). However, they also mention the use of granites and other volcanics. This is a bit concerning and something that must be addressed, because stone raw materials matter so much to tool selection. This is especially because the main differences identified between the two study sites is intensity of tool use. Granites and limestones will have very different mechanical properties. There could be a scenario whereby the differences in raw material structured the degree of utilization because some rocks were softer and would need to be utilized further. However, if the proportions of different lithologies were the same on both islands then this would be a moot point. The authors might also reconsider use of the word "traction" when they referred to the process by which these non-limestone rocks made it to the island. Do they mean here that they were brought to the coast as cobbles/pebbles by wave action, but are not actually lithic materials found on the islands themselves as outcrops?

8) Clarification of the tidal and intertidal significance. There needs to be a little bit better explanation as to why the intertidal versus the tidal zone matters. It is clear that the authors went out of their way to document this in great detail, but it is not quite clear why. Are there ecological differences between these zones that might impact the intensity of use of the resources in them? There do appear to be compelling differences in the variability of rock size availability at the two different locales, but why does this matter in terms of which rocks would be selected by macaques?

9) Application to the archaeological record in the discussion and conclusions. The conclusion mentions there are few ways of testing cultural transmission in the archaeological record. However, this is technically also true of this study. What the authors have documented is differences that archaeologists usually identify as functional (degree of utilization). Several studies have focused on degree of utilization as measures of site use intensity (e.g. Dibble, 1995 in Dibble and Lenoir volume). However, as the authors here have argued, these differences may be cultural IF we can determine that ecological factors and genetic factors can be ruled out. The genetic factors will unlikely be identified in the archaeological record. However, the contextual ecological factors may be identified. The one caveat with the Luncz et al. is that these are groups on islands. Might we expect similar kinds of diversity in Paleolithic contexts whereby information can pass between groups more easily? In Paleolithic contexts the assemblages are heavily time averaged. Given what Luncz and colleagues have already shown with resource depression that tool use can begin to wane because the prey items decrease. What are the possible time depths of the behaviors that macaques are exhibiting now? What would the time-averaged signature of the behaviors on Lobi Bay look like? Some discussion of this would make the paper even more archaeologically applicable.

Title

The title should reflect the new emphasis on the application to archaeology.

---

## [Author Response]

Essential revisionsBelow is a summary of the key changes that should take place to make this manuscript acceptable for publication in eLife:1) Clear integration of this research with the previous study. As a Research Advance, a more substantial review of the results of the previous study would be beneficial. How and why are these studies different and how are they alike? It is not clear enough to readers who are unfamiliar with macaque tool use sites that the two studies were conducted with two entirely separate populations at different sites. Also, since the original study found that ecological parameters do appear to be driving some of the variation in tool use, the deviation from this in the current study becomes even more interesting. The previous study capitalized upon the size differential between the two macaque populations to examine whether habitual tool-aided exploitation by macaques could lead to reduction in size and abundance of shellfish prey. This contributed to the long-standing debate in coastal archaeology as to whether the reduction in shellfish sizes in the Middle Stone Age could have been due to human exploitation of marine resources. Here, this study is showing that primate tool artifacts also vary between populations. The authors should emphasize these potential contributions via primate archaeology, and how further behavioral work could help with interpreting the lithic record (particularly with respect to the concerns raised above).

We have expanded the review of our previous study and further highlighted the importance of our new findings.

2) Re-framing the significance of the work. This should be with respect to what is directly observable, and more specifically address its implications for the archaeological record of early hominins, rather than simply labelling patterns as cultural or non-cultural.

We have reframed the significance of this work in regard to the implications for the archaeological record.

3) Clarifying possible alternative explanations than "culture". The lack of information about the sex and age of individuals does not invalidate this study, and in fact makes it more akin to archaeological work. However, all possible sources of variation should be mentioned as possible explanations. One possibility is providing information on the number of individuals or different age on the two islands. If the islands do not differ in their age and sex demographics then it would seem unlikely that this is a reason for these differences. However, the authors should still discuss other possibilities. How confidently can we disentangle the effects of use frequency from users' strength/proficiency/strike accuracy on wear patterns?

We have included additional information on the demographics of the studied populations and highlighted possible alternative explanations for the observed differences in tool variation in regard to the primate archaeological record.

Macaques do differ in proficiency, and some tool users will be able to crack food items with fewer strikes or might be more accurate with their strikes. These users would conceivably create fewer and less diffuse use wear than users who require many strikes to crack food items. The latter might create use wear that would be scored as more intense or extensive but interpreting this as re-use would be erroneous. Perhaps differences in use wear patterns between both populations could indicate differences in population level proficiency? In spite of the authors' claim that the snails are morphologically similar, there could be subtle ecological differences that could explain a different use of stones by the monkeys on the two islands. For instance, there might be a higher proportion of a certain type of snail in one of the two islands, which would make the re-use of certain stones more efficient.

The amount to tools for each prey species that we found was approximately similar. In order to identify differences in use wear intensity we applied a modified use wear grading system published by Haslam et al., 2013. This system was initially developed to assign the behaviour to the tool but not to assess the intensity of use. We added additional clarification to the Materials and methods section of the paper to address this methodology.

There is also a cognitive argument that can be made about tool selection, e.g. functional fixedness. Macaques on Boi Yai have to use the heavier tools to crack oysters because they are bigger than on the other island. That means the heavier tools can be functionally fixed to be used with oysters. Macaques then use lighter rocks (the ones that are not functionally fixed) to crack the snails. On the other hand, on Lobi Bay, there is no functional fixedness at work, and there is an additional problem: stones appear to be rather lighter in general, in comparison to the ones in Boi Yai, at least according to Figure 8: it is unclear is a rock of less than 100g is enough to break a snail, so macaques could be drawn there to take heavier stones.

We highlight in the text that tool selection is partly influenced by the prey size. This is consistent with other studies which show that tool size is adjusted to prey size. However this fact holds true for both populations. Raw material availability however is not a limiting factor for tool selection. We added more details regarding this to the text but highlight the fact that this factor does not explain all differences found in their tool evidence.

On that note, there is confusion about the difference in weight that can be found in the figures. In Figure 8, weight ranges from less than 100 to around 200g. On Figure 5, weights are rather around 400-800g. If these are estimates obtained from the model, the authors should say it.

The plot is based on the original data. Figure 9 is not showing the range of the original data, only the 95% confidence interval. We have clarified this in the text.

4) More in-depth treatment of the archaeological and primate archaeological literature. The mention of pounding tools as part of the toolkit of the LCA could likely include references to Rolian and Carvalho, 2017. The authors should also reference John Shea's recent paper (Evol Anthropology, 2018) about tool use as habitual versus occasional, and perhaps Thompson's (2019 Current Anthropology) piece on pounding tool use as the precursor to flaked stone tool use.

We have included the suggested literature.

Elsewhere in the Introduction the authors mention the idea of the extraordinary advantages that tool use provides those individuals that implement these strategies. While this makes sense to me theoretically, there are only a few examples where this has been explicitly tested. One example would be Morgan, 2016 who explicitly tested the significance of tools. Similarly, the final sentence of paragraph four on the Introduction that mentions that fact that primate tool use leaves an archaeological record could benefit from references to Carvalho et al., 2012. This would however call for a review of other papers that have been published on the topic, and what this particular paper brings in addition.

We have deleted this statement.

5) Clarification and de-emphasis of multiple use tools (or removal of this argument). The argument on multitool or curated (in the sense used by the authors here, but see point 6 below) usage is, right now, too weak. The idea of multi-use tools is interesting, but cannot be used here as one of the main arguments for differences between groups. How do we know it was used on multiple prey-species? The authors state that if a specimen could not be identified as being associated with a single species then it was not included in the study, so this would contradict that. Furthermore, there are only 7 stones observed, and the researchers haven't been able to observe that they were really used for different purposes. In general, the discussion of the Boi Yai island tool is problematic (L298+). Following the argument of the authors, there is not much use wear for snails because little force is needed. But then, the same argument should apply for oysters: because they are bigger, then more force should apply and thus likelihood of use wear increases. And once again, functional fixedness or conservatism, which are both mechanisms acting at the individual (rather than 'cultural') level, can kick in, and explain the re-use, because it is unclear what macaques consider the same substrate and what is not. In fact, it is unclear whether re-use is evidence of culture in any animal species so far (see for example Hobaiter et al., 2014, Plos Biology), rightly so because of the possibility of individual characteristics (it is much more parsimonious than to claim norms for instance). In addition, if some tools were indeed reused, how can the authors be sure that the findings are not contributed by single or several individuals? The authors do not have behavioral data from this population to confirm this, but observations of tool use by macaques at other sites should make them aware that some individuals more than others have stronger tool preference and retain tools for longer periods of time than others. This possibility must at least be discussed. The authors interpret scarring on different tool surfaces as a repurposing of the tool for different food items. This is based on previous findings on one population of macaques on Piak Nam Yai island that tool points were predominantly used on oysters while faces were used on gastropods (Haslam et al., 2013, Tan et al., 2015). More current research on yet another population on Koram Island however, is finding that the macaques there predominantly use tool points regardless of whether or not they were processing sessile oysters or motile gastropods. This work is preparation and not yet published, so it is understandable that the authors not consider this in their interpretations of use wear at present, but it does signal a need for greater caution.

We have removed the multi-use tool argument from the manuscript.

A final possibility might be that the difference in use intensity between two populations is related to differences in tidal conditions. From Figure 1, it appears that the Boi Island site is relatively more sheltered from the open sea compared to Lobi Bay. We know tool-aided coastal foraging can take place only during the low tides, and used tools left along the shore can be washed away by wave action. Is it possible that the macaques on Boi Island have greater access to the coast, and that tools on Boi Island are less likely to be washed away by wave action?

This is unlikely, Lobi Bay (as it lies within a bay) is essentially more sheltered than Boi Yai island. Further research will test for the duration of tool evidence along the shore. We included this discussion into the manuscript.

6) Clarification of terminology to connect archaeology and animal behavior literatures. The authors use the word "curation" several times in the manuscript. This is an interesting use for the word in this context, but the authors need to be more explicit about what they are describing. The word curation in relation to tool use stems from Lewis Binford's investigation into the variance in resource and land use by modern hunter-gatherers and his gatherer vs. forager dichotomy (Binfrod, 1973; 1979; 1980). However, largely stemming from the vague nature of this description, Shott and others have reviewed this concept. The most current concept of curation in archaeological contexts is the realized utility of a tool relative to the actual utility of a tool. This is after an extensive review of what is meant by curation by Shott (1996 as well as subsequent reviews of use life Shott and Sillitoe, 2006). It would be exciting to see primate archaeology begin to use the vocabulary of Paleolithic archaeologists in this way, but if the authors intend to do so they should describe what they mean by curation and implement it into this manuscript (with references to the original literature). If they are not going to do that, then they should use a different vocabulary altogether. For example, the term "repeated curation" is not how archaeologists would use this term, and curation would not be something that is repeated. This would conflate curation with re-use. If they use the term in the way that Shott describes it they could quite easily accommodate the idea that the Boi Ya macaques are extracting a greater amount of utility from the tools then those at Lobi Bay. In archaeological literature this would be described as differing levels of use life and site use intensity usually associated with ecological differences. However, I think the authors have convincingly shown that this is unlikely, especially since the greater levels of intensity of use are not seen in all species. Another example of terminology that does not connect the two bodies of literature is in how the authors describe transects as "background material" which is used to describe the locally available stones that were not selected for tool use. It is excellent that they are providing the data for this, which is directly comparable to archaeological data which do the same thing. However, in that case it should be referred to as "locally available raw material" as opposed to "background material" because it will make conversations between archaeologists and primate archaeologists more productive to be using the same vocabulary.

We have added the suggested literature and discussed how curation could be applied to primate archaeology.

7) Description of raw material. The description of the rock types used by the macaques indicates that they use mostly limestone but also use calcite (which seems like a very bad option for cracking shells but I guess it is what you need to use if you have no other options). However, they also mention the use of granites and other volcanics. This is a bit concerning and something that must be addressed, because stone raw materials matter so much to tool selection. This is especially because the main differences identified between the two study sites is intensity of tool use. Granites and limestones will have very different mechanical properties. There could be a scenario whereby the differences in raw material structured the degree of utilization because some rocks were softer and would need to be utilized further. However, if the proportions of different lithologies were the same on both islands then this would be a moot point. The authors might also reconsider use of the word "traction" when they referred to the process by which these non-limestone rocks made it to the island. Do they mean here that they were brought to the coast as cobbles/pebbles by wave action, but are not actually lithic materials found on the islands themselves as outcrops?

We have modified the text of the manuscript to make explicit that there is no calcite included in this study. Additionally, we have expanded Table 1. to show the frequencies of hammerstones of each raw material for both islands. Granite hammerstones only account for a single artefact and as such does not have an effect on the statistical analysis presented in this study. We added a short discussion regarding how granite material might have appeared at the field sites.

8) Clarification of the tidal and intertidal significance. There needs to be a little bit better explanation as to why the intertidal versus the tidal zone matters. It is clear that the authors went out of their way to document this in great detail, but it is not quite clear why. Are there ecological differences between these zones that might impact the intensity of use of the resources in them? There do appear to be compelling differences in the variability of rock size availability at the two different locales, but why does this matter in terms of which rocks would be selected by macaques?

We have clarified this in the text.

9) Application to the archaeological record in the discussion and conclusions. The conclusion mentions there are few ways of testing cultural transmission in the archaeological record. However, this is technically also true of this study. What the authors have documented is differences that archaeologists usually identify as functional (degree of utilization). Several studies have focused on degree of utilization as measures of site use intensity (e.g. Dibble, 1995 in Dibble and Lenoir volume). However, as the authors here have argued, these differences may be cultural IF we can determine that ecological factors and genetic factors can be ruled out. The genetic factors will unlikely be identified in the archaeological record. However, the contextual ecological factors may be identified. The one caveat with the Luncz et al. is that these are groups on islands. Might we expect similar kinds of diversity in Paleolithic contexts whereby information can pass between groups more easily? In Paleolithic contexts the assemblages are heavily time averaged. Given what Luncz and colleagues have already shown with resource depression that tool use can begin to wane because the prey items decrease. What are the possible time depths of the behaviors that macaques are exhibiting now? What would the time-averaged signature of the behaviors on Lobi Bay look like? Some discussion of this would make the paper even more archaeologically applicable.

We have added this point of discussion to the manuscript.

TitleThe title should reflect the new emphasis on the application to archaeology.

We have adjusted the title to reflect a more archaeological take on the data.